# The Role of the Histone Methyltransferase EZH2 in Liver Inflammation and Fibrosis in STAM NASH Mice

**DOI:** 10.3390/biology9050093

**Published:** 2020-05-02

**Authors:** Seul Lee, Dong-Cheol Woo, Jeeheon Kang, Moonjin Ra, Ki Hyun Kim, Seoung Rak Lee, Dong Kyu Choi, Heejin Lee, Ki Bum Hong, Sang-Hyun Min, Yongjun Lee, Ji Hoon Yu

**Affiliations:** 1New Drug Development Center, Daegu-Gyeongbuk Medical Innovation Foundation, Daegu 41061, Korea; autrition15@dgmif.re.kr (S.L.); dongkyu@dgmif.re.kr (D.K.C.); jini150117@gmail.com (H.L.); kbhong@dgmif.re.kr (K.B.H.); shmin03@dgmif.re.kr (S.-H.M.); 2Convergence medicine research center, Asan Institute for Life Sciences, Asan Medical Center, and Department of Convergence Medicine, University of Ulsan College of Medicine, Asan Medical Center, Seoul 05505, Korea; dcwoo@amc.seoul.kr; 3Center for Bio-Imaging of New Drug Development, Asan Life Science Institution, Asan Medical Centre, Seoul 05505, Korea; ibelieveinmiracle@gmail.com; 4Hongcheon Institute of Medicinal Herb, 101 Yeonbongri, Hongcheon 25142, Korea; ramj90@himh.re.kr; 5School of Pharmacy, Sungkyunkwan University, Suwon 440-746, Korea; khkim83@skku.edu (K.H.K.); davidseoungrak@gmail.com (S.R.L.); 6School of Life Sciences and Biotechnology, BK21 Plus KNU Creative BioResearch Group, Kyungpook National University, Daegu 41566, Korea

**Keywords:** EZH2, enhancer of zeste homolog 2, H3K27me3, trimethylation on Lys 27 of histone H3, histone methyltransferase (HMT)

## Abstract

Non-alcoholic fatty liver disease (NAFLD) is a leading form of chronic liver disease, with few biomarkers and treatment options currently available. Non-alcoholic steatohepatitis (NASH), a progressive disease of NAFLD, may lead to fibrosis, cirrhosis, and hepatocellular carcinoma. Epigenetic modification can contribute to the progression of NAFLD causing non-alcoholic steatohepatitis (NASH), in which the exact role of epigenetics remains poorly understood. To identify potential therapeutics for NASH, we tested small-molecule inhibitors of the epigenetic target histone methyltransferase EZH2, Tazemetostat (EPZ-6438), and UNC1999 in STAM NASH mice. The results demonstrate that treatment with EZH2 inhibitors decreased serum TNF-alpha in NASH. In this study, we investigated that inhibition of EZH2 reduced mRNA expression of inflammatory cytokines and fibrosis markers in NASH mice. In conclusion, these results suggest that EZH2 may present a promising therapeutic target in the treatment of NASH.

## 1. Introduction

NAFLD is a complex spectrum of chronic liver diseases ranging from steatosis to nonalcoholic steatohepatitis (NASH), an advanced form with hepatocyte inflammation [1,2,3,4]. NASH is characterized by histological lobular inflammation and hepatocyte ballooning and fibrosis [5]. Patients with NASH progress cirrhosis and hepatocellular carcinoma (HCC) have an increased risk of liver-related patient death compared to simple steatosis patients [6]. Although the progressive mechanism of NASH is not well understood, NASH-related target proteins have been identified in various cases of obesity, diabetes, and hyperlipidemia [7]. Indeed, several therapeutic strategies targeting the development of steatosis and steatohepatitis have been evaluated [8]. However, according to the FDA, there is no registered treatment for NASH, yet obeticholic acid, selonsertib, elafibranor, and cenicriviroc have entered phase III stage development [9]. Currently, we need to find therapeutics because there are no FDA-approved therapies for NASH. Vitamin E and pioglitazone as traditional therapies have effect on steatosis and inflammation. However, treatment of Vitamin E and pioglitazone does not reduce liver fibrosis as a strong marker of NASH [10]. Epigenetic modification may control the development of NAFLD causing non-alcoholic steatohepatitis (NASH) [11,12]. Despite evidence that DNA methylation may contribute to the progression of NAFLD [13,14], the role of histone modifications in NASH have yet to be explored.

Enhancer of Zeste Homolog 2 (EZH2) as a histone methyltransferase (HMT) regulates mono- through trimethylation of histone 3 at lysine 27 (H3K27). H3K27 trimethylation (H3K27me3) by EZH2 is a key mechanism responsible for gene repression [15]. EZH2 is highly expressed in solid tumors and is involved in poor prognosis in hepatocellular carcinoma (HCC) [16].

In various mouse models, the NASH model has developed the genetic leptin-deficient mouse (ob/ob) or dietary methionine-choline–deficient (MCD) mouse. However, these mouse models do not reflect human NASH phenotype and pathogenically mechanisms [17].

In the STAM NASH model, streptozotocin (STZ) is administered to produce inflammation, impairment of insulin secretion, and elucidate the resultant phenotype for advanced type 2 diabetes (T2D) [18]. The NASH STAM mouse model is widely used because it presents the full spectrum of human NAFLD ranging from steatosis to fibrosis. The histological phenotype of the STAM NASH model has been observed to be similar to the phenotype from human clinical samples [19].

STAM mice develop non-alcoholic steatohepatitis (NASH) at eight weeks, which leads to fibrosis at 12 weeks, and finally to HCC in males [20]. Therefore, the NASH STAM model was used to investigate the wide spectrum of NAFLD, ranging from steatosis to non-alcoholic steatohepatitis (NASH) and liver fibrosis. Epigenetic regulation mechanisms involved in the development and progression of NASH have been determined yet. Therefore, we have investigated the role of EZH2 in an in vivo NASH model.

## 2. Material and Methods

### 2.1. Animals and Experimental Design

Pregnant C57BL/6 mice were obtained from Dae-Han Biolink Co., Ltd. (Eumsung, Korea) and two-day-old male pups were injected with streptozotocin (200 μg per mouse) and fed a high-fat diet from the age of four weeks on. The STATM mouse model progressed from NAFLD to non-alcoholic steatohepatitis (NASH) at eight weeks of age. In the NASH-targeting study, obeticholic acid, EPZ-6438, and UNC1999 were administered orally once daily at 10 mg/kg from six weeks to nine weeks. Mice were euthanized by cardiac exsanguination under general anesthesia. Blood and liver samples were collected for NASH-related marker detection and real-time polymerase chain reaction analysis. All protocols and procedures conformed to the guidelines of the Hongcheon Institute of Medicinal Herb Committee for Care and Use of Laboratory Animals and were approved by the Animal Experiments Ethics Committee of Hongcheon Institute of Medicinal Herb.

### 2.2. Treatment Groups

For this study, EZH2 inhibitors EPZ-6438 and UNC1999 were administered orally once daily at 10 mg/kg over three weeks. For positive control, Obeticholic acid, an FXR agonist, was purchased from MedChemExpress. The stock solution of Obeticholic acid was dosed orally once a day at 10 mg/kg over three weeks.

### 2.3. Body Weight

The mice were weighed daily between 15:00 and 17:00.

### 2.4. Histological Analysis

Mouse liver tissues were removed and fixed in 10% neutral buffered formalin and embedded in paraffin wax. Then, 5-μm sections were prepared for hematoxylin and eosin (H&E) staining and stained with Oil Red O to illustrate hepatic lipid accumulation.

### 2.5. Evaluation of Biochemical Parameters

The animals were anesthetized, after which 500 μl of blood was collected. The serum was separated from blood and used for tumor necrosis factor-α (TNF-α) and alanine aminotransferase (ALT) assays.

### 2.6. Real-Time Polymerase Chain Reaction Analysis

Total RNA was extracted from the liver using an RNeasy Mini Kit (Qiagen 74104, Gaithersburg, MD, USA) and reverse transcribed using a High Capacity cDNA Reverse Transcription kit (Applied Biosystems, Foster City, CA, USA). The cDNA was amplified using a master mix purchased from Applied Biosystems, followed by amplification at 40 cycles of denaturation and 95 °C for 15 s, annealing at 55 °C to 60 °C for 15 s, and extension at 72 °C for 30 s. All samples were normalized to the expression of GAPDH. The cDNA was used for PCR with specific primers for Desmin, DGAT1, IL-6, TGF-β, IL-1β, LCAD, CTGF, TNF-α, and GAPDH. The following primers were used for QRT-PCR analysis: mouse GAPDH: forward 5′-AACGACCCCTTCATTGAC-3′, and reverse 5′- TCCACGACATACTCAGCA-3′; mouse Desmin: forward 5′- TCAGCGAGGCTACACAGCAACA-3′, and reverse 5′-GGTTGGGCAGCATGAAGACCACAA -3′; mouse Timp-1: forward 5′- TGCCTGCTGCGATTACAACC -3′, and reverse 5′- GGAATGGTGTGGTGATGCATGG -3; mouse DGAT1: forward 5′- TCCGCCTCTGGGCATTC -3′, and reverse 5′- GAATCGGCCCACAATCCA-3′, mouse IL-6: forward 5′- AGTTGCCTTCTTGGGACTGA -3′, and reverse 5′- TCCACGATTTCCCAGAGAAC -3′; mouse TGF-β: forward 5′- TTGCCCTCTACAACCAACACAA-3′, and reverse 5′- GGCTTGCGACCCACGTAGTA -3′; mouse IL-1β: forward 5′- TCAGGCAGGCAGTATCACTCATT-3′, and reverse 5′- GGAAGGTCCACGGGAAAGA -3′; mouse LCAD: forward 5′- CAGAGAAACATGGCGGCA -3′, and reverse 5′- AGCCAGCGCGTGTGCAATT -3′; mouse CTGF: forward 5′- AGAACTGTGTACGGAGCGTG-3′, and reverse 5′- GTGCACCATCTTTGGCAGTG -3′; mouse TNF-α: forward 5′- CGTGCTCCTCACCCACAC-3′, and reverse 5′- GGGTTCATACCAGGGTTTGA -3′; mouse EZH2: forward 5′-CTGCTGGCACCGTCTGA-3′, and reverse 5′-GTTGCATCCACCACAAA-3′; mouse IFN-gamma: forward 5′- TGAGCTCATTGAATGCTTGG-3′, and reverse 5′-ACAGCAAGGCGAAAAAGGAT-3′; mouse Runx3: forward 5′-GGTTCAACGACCTTCGATTC-3′, and reverse 5′-CGGTGGTAGGTAGCCACTTG-3′. The expression of gene may be measured by relative quantification using the comparative Ct method. This method measures the Ct difference between the target gene and the internal reference gene (18S) and then compares the ΔCt values of treated samples to control group samples.

### 2.7. Statistical Analysis

The statistical differences were determined one-way analysis of variance (ANOVA) and Newman–Keul’s test. All values are expressed as means ± standard error (SE), and statistical significance were set at *p* < 0.05.

## 3. Results

### 3.1. EZH2 Inhibitors Reduce Liver Steatosis in NASH Mice

We aimed to determine whether treatment of EZH2 inhibitors may be linked to changes in liver steatosis. We performed small molecule UNC1999 and EPZ6438 as EZH2 inhibitors and obeticholic acid as clinical trial III compound for NASH treatment in STAM mice to evaluate its efficacy on liver steatosis. The STAM mice produced valuable information on monitoring the progression from steatosis to NASH. Histological analysis evaluated that STAM mice had liver steatosis, lobular inflammation, hepatocyte ballooning at six weeks and had fibrosis at 10 weeks with the start of the high-fat diet at four weeks [21]. The STAM mice were orally dosed once a day with EZH2 inhibitors (10 mg/kg) for three weeks, starting at six weeks of age.

The Obeticholic acid (10 mg/kg), an FXR agonist, was included as a positive control in the studies. We divided five groups including normal group (Healthy mice), control group (NASH STAM mice), obeticholic acid group (NASH STAM mice + obeticholic acid), UNC1999 group (NASH STAM mice + UNC1999 EZH2 inhibitor), and EPZ6438 (NASH STAM mice + EPZ6438 EZH2 inhibitor). We analyzed liver damage and steatosis in mouse liver tissue (Figure 1B,C). Using Oil Red O staining, liver steatosis showed a decrease in group of treatment with EZH2 inhibitors and obeticholic acid in the studies. The steatosis in NASH mice was attenuated by treatment with UNC1999 and EPZ6438 as EZH2 inhibitors and obeticholic acid (Figure 1C). EZH2 inhibition for three weeks with inhibitors in STAM mice attenuated liver steatosis indicated by H&E (Figure 1B) and Oil Red O (Figure 1C) staining of liver sections. Therefore, treatments with EZH2 inhibitors reduced liver fat accumulation in tissues.

### 3.2. There is No Effect of EZH2 Inhibitors on Body Weight

We aimed to determine whether the effects of EZH2 inhibitors may be linked to changes in body weight. We also examined whether EZH2 inhibitors had specific target organ toxicity. There was no difference between STAM mice + EZH2 inhibitors group and STAM mice (control) for body weight (Figure 2A). Furthermore, there were no significant between-group differences regarding the liver, kidney, and white adipose tissue relative weight (Figure 2B). Therefore, treatments with EZH2 inhibitors were well tolerated in the STAM mice and measured no deleterious effects on body weight.

### 3.3. EZH2 Inhibitors Have No Protective Effect against Liver Inflammatory Markers TNF-A, ALT in NASH Mice Serum

We aimed to determine whether the effect of EZH2 inhibitors on liver inflammation may be explained by markers changes in mouse serum. To explore this hypothesis, we tested small molecule EZH2 inhibitors, UNC1999 and EPZ6438, in STAM mice to evaluate its efficacy on NASH. The STAM mice produced valuable information on monitoring the progression from steatosis to NASH. The STAM mice were orally dosed once a day with EZH2 inhibitors (10 mg/kg) for three weeks, starting at six weeks of age. The Obeticholic acid (10 mg/kg), an FXR agonist, was included as a potential inhibitor in the NASH study. TNF-α, ALT, and glucose are the major markers of liver inflammation and injury caused by non-alcoholic fatty liver disease (NAFLD). We investigated whether EZH2 inhibitors affected the serum levels of TNF-α in mouse serum. The TNF-α level was slightly lower in the STAM mice + UNC1999 group than in the STAM mice (control) group. However, there was no effect on TNF-alpha after treatment with EPZ6438 (Figure 3A). Also, there was no change the level of ALT and glucose between STAM mice treated with EZH2 inhibitors and STAM mice controls (Figure 3B).

### 3.4. EZH2 Inhibitors Regulate the Expression of EZH2 Direct Target and Inflammatory Related Genes

We analyzed gene expression of EZH2 gene, EZH2 relative target Runx3 gene, and inflammation marker IFN-γ gene in mouse liver tissue (Figure 4). Increasing EZH2 mRNA expression in NASH STAM mice showed a significant decrease in EZH2 mRNA expression following treatment with EZH2 inhibitors. In lung fibroblast LL29 and chondrosarcoma cells, treatment of 3-Deazaneplanosin A (DZNep), EZH2 inhibitor reduces EZH2 protein expression [22,23]. Runx3 is well known EZH2 target gene. Treatment of EZH2 inhibitors induced Runx3 mRNA expression in tissues because EZH2 has a key mechanism responsible for gene repression (Figure 4). Treatment of EZH2 inhibitors reduced inflammatory cytokines IFN-γ mRNA expression in tissues (Figure 4).

### 3.5. EZH2 Inhibitors Reduce the Expression of Inflammatory and Fibrosis Related Genes

We analyzed gene expression of the inflammatory cytokines interleukin-1 beta (IL-1β), interleukin-6 (IL-6), and the fibrosis markers, connective tissue growth factor (CTGF), transforming growth factor-β (TGF-β), diacylglycerol acyltransferase 1 (DGAT1), and diacylglycerol acyltransferase 1 (LCAD) in mouse liver tissue (Figure 5). Several markers showed a significant decrease in inflammation and fibrosis-related mRNA expression following treatment with EZH2 inhibitors in the studies.

## 4. Discussion

Epigenetic alterations caused by the environment, genetic factors, and hypercaloric diets may regulate the gene expression leading to the progression and development of NAFLD [10,11]. In the epigenetic mechanism of chronic liver diseases, hepatic methylations are positively correlated with the severity of NAFLD. Indeed, DNA methyltransferase 1 (DNMT1) was upregulated in NASH patients [24]. Also, the deletion of HDAC3 in mice results in steatosis, inflammation and fibrosis [25].

Histone methyltransferase EZH2 is overexpressed in various malignant tumors and involved in promoting tumor growth and metastasis [26,27,28]. Recently, it has been suggested that EZH2 promotes fibrosis in the lungs, liver, and kidneys [22,29,30]. However, the relationship between NASH and EZH2 is not well understood.

In this study, we evaluated the physiological and molecular functions and regulation of EZH2 in NASH pathogenesis, showing that treatment of EZH2 inhibitors EPZ6438, UNC1999 in an in vivo model of STAM NASH mice. It is the first study in which the inhibition of the EZH2 has been demonstrated to reduce liver inflammation and fibrosis in NASH mice.

EPZ6438 (Tazemetostat) is also an orally bioavailable small molecule inhibitor with high affinity and selectivity for EZH2 [31]. EPZ6438 has selectivity for EZH2 with 35-fold potency compare to EZH1 and over 4500-fold the potency of other histone methyltransferases. In phase II clinical trials, EPZ6438 is being tested in Diffuse Large B Cell Lymphoma. UNC1999 is the first orally bioavailable inhibitor with high potency for EZH2 wild-type and Y641 mutant enzymes with IC50 of 2 nM in in vitro enzyme assays [32]. This inhibitor suppresses H3K27 trimethylation levels in cells and selectively inhibits both B cell lymphoma cell lines and growth of mixed lineage leukemia (MLL)-rearranged leukemia cells [33]. According to Vella et al., the depletion of EZH2 exacerbates inflammation in NAFLD [34]. EZH2 expression decreased in high-fat/high-fructose diet (HFa/HFr-D) in vivo NAFLD model and palmitic/oleic acid (PA/OA) in vitro lipid accumulation HepG2 model. Treatment of DZNep as EZH2 inhibitor induced lipid accumulation in PA/OA induce in vitro HepG2 model. However, we suggest that opposite effect of EZH2 expression and EZH2 inhibitor might cause NASH model difference between HFa/HFr-D induced NAFLD and streptozotocin (STZ)/high-fat induced NASH model.

## 5. Conclusions

Here, we determined that treatment of EZH2 inhibitors induced anti-inflammatory effects as evidenced by a reducing the expression levels of IL-6, IL-1β, IFN-γ, TGFβ, and CTGF for inflammation and fibrosis markers in liver tissues. In conclusion, the results from our study suggest that EZH2 inhibitors may represent a promising therapeutic in the treatment of NASH.

## Figures and Tables

**Figure 1 biology-09-00093-f001:**
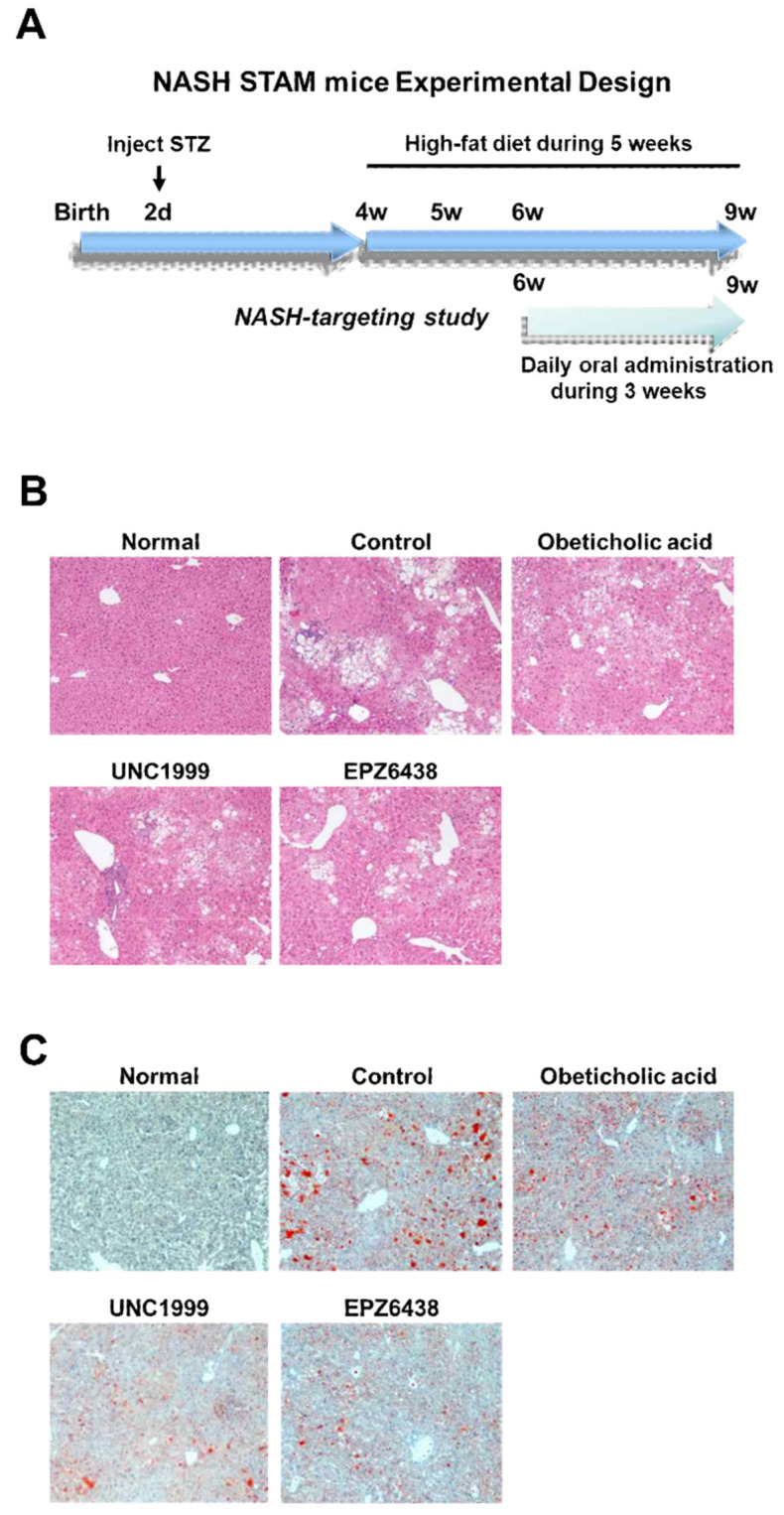
Treatment with Enhancer of Zeste Homolog 2 (EZH2) inhibitors reduces liver steatosis in STAM NASH mice. (**A**) NASH STAM mice experimental design. STAM-Vehicle (Control) were injected with streptozotocin (STZ) on day 2 to induce a diabetic state. Mice were fed a high-fat diet from four weeks. Mice were dosed with either vehicle, Obeticholic acid, UNC1999, or EPZ6438 once a day from 6–9 weeks for the study. (**B**) Representative hematoxylin and eosin (H&E) stained (200x) liver sections from healthy (Normal), STAM-Vehicle (Control), STAM-Obeticholic acid (Obeticholic acid), STAM-UNC1999 (UNC1999), and STAM-EZH2 (EPZ6438) groups in the NASH mice. (**C**) Representative Oil Red O stained (200x) liver sections from healthy (Normal), STAM-Vehicle (Control), STAM-Obeticholic acid (Obeticholic acid), STAM-UNC1999 (UNC1999), and STAM-EZH2 (EPZ6438) groups in the NASH mice. Healthy (Normal), n = 5; Vehicle (Control), n = 5; STAM-Obeticholic acid (Obeticholic acid), n = 5; STAM-UNC1999 (UNC1999) n = 5; STAM-EZH2 (EPZ6438), n = 5.

**Figure 2 biology-09-00093-f002:**
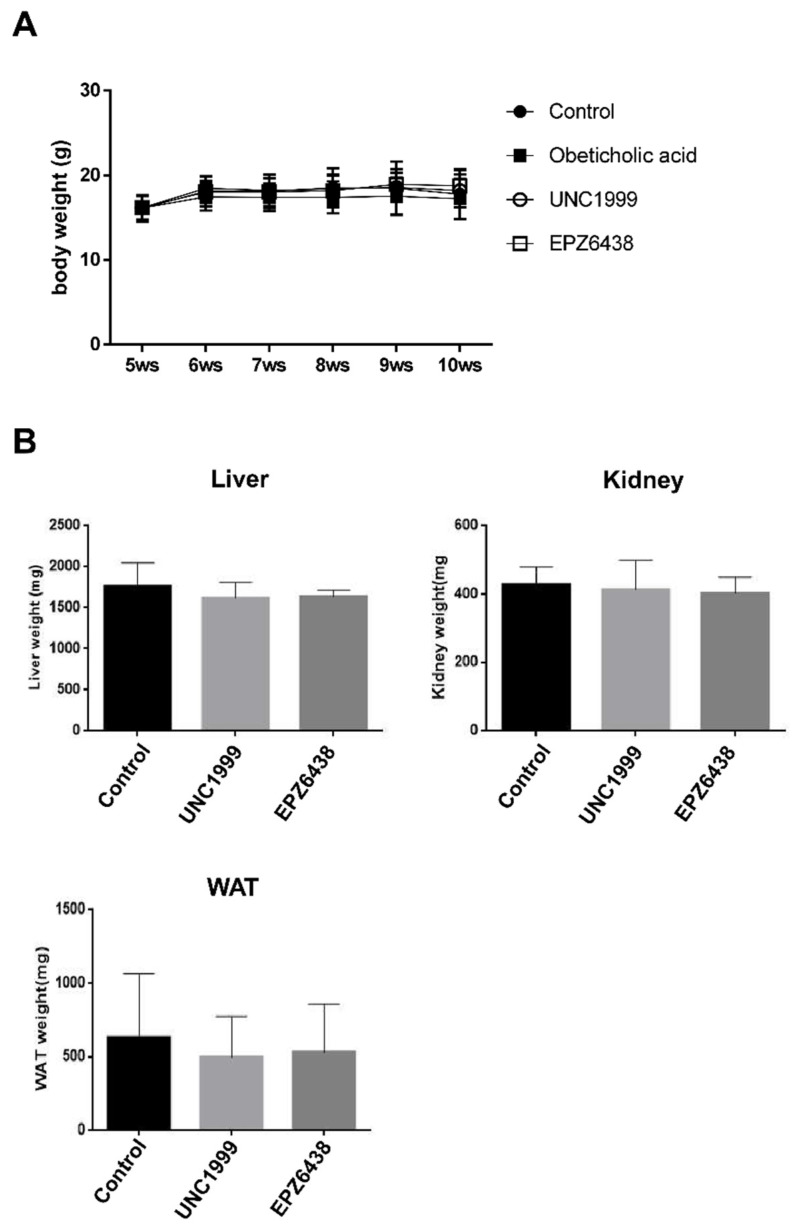
Treatment with EZH2 inhibitors has no effect on body weight in STAM NASH mice. (**A**) Effect of EZH2 inhibitors (UNC1999, EPZ-6438) and obeticholic acid on body weight. (**B**) Effect of EZH2 inhibitors on the liver, kidney, and white adipose tissue relative weight. Relative organ weight was measured as the weight of the organ divided by the body weight. Data are shown as mean ± standard deviation (n = 5). Abbreviations: WAT, white adipose tissue.

**Figure 3 biology-09-00093-f003:**
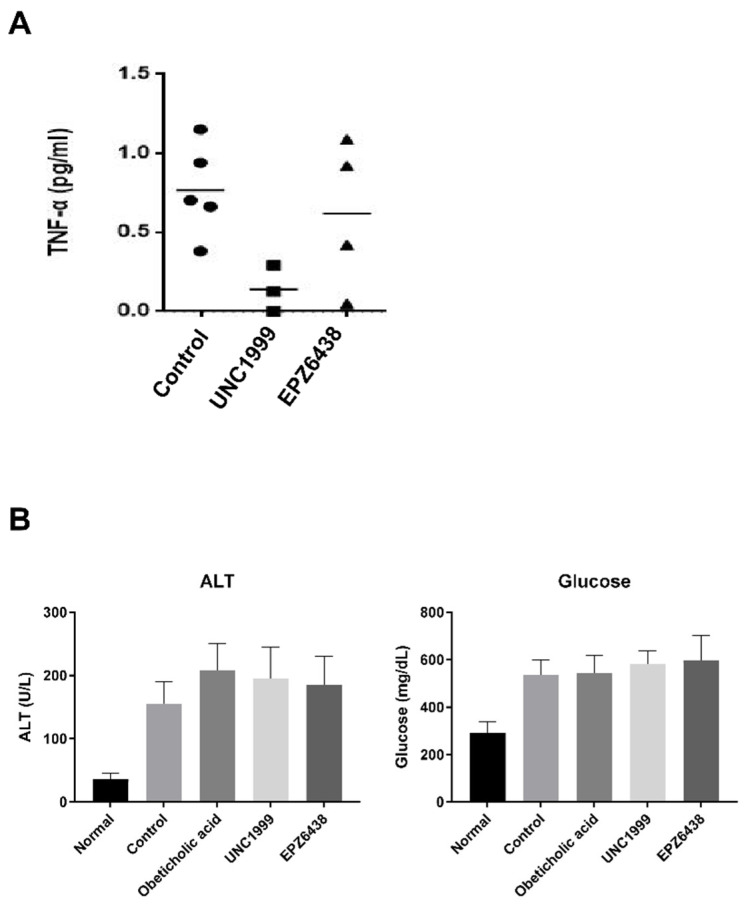
Treatment with EZH2 inhibitors attenuates liver inflammation in STAM NASH mice. (**A**) Effect of EZH2 inhibitors (UNC1999, EPZ-6438) treatment on the serum levels of tumor necrosis factor-α (TNF-α). (**B**) Effects of EZH2 inhibitors and obeticholic acid treatment on the serum levels of alanine aminotransferase (ALT) and glucose. Statistically significant differences versus NASH group and EZH2 inhibitors treatment group are marked as * *p* < 0.05, respectively.

**Figure 4 biology-09-00093-f004:**
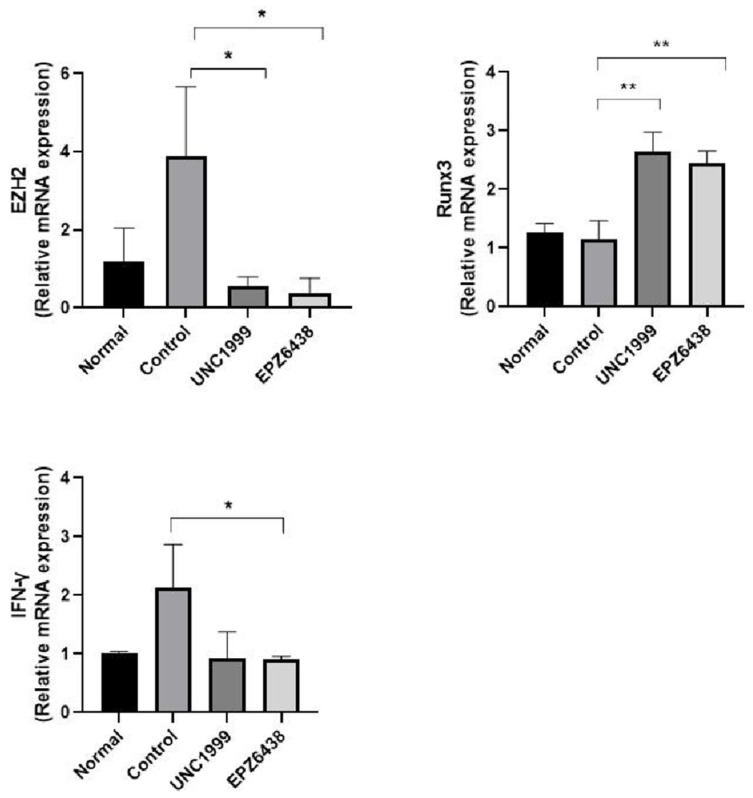
Treatment with EZH2 inhibitors inhibits ezh2 target Runx3 gene and liver inflammation-related IFN-γ gene in STAM NASH mice. Gene expression of EZH2 target and inflammatory marker from Real-Time PCR. Expression of genes in healthy (Normal), STAM-Vehicle (Control), STAM-UNC1999 (UNC1999), and STAM-EZH2 (EPZ6438) groups in the NASH mice. The final concentration of UNC1999 and EPZ6438 is 10 mg/kg. Statistically significant differences versus the STAM-Vehicle group and EZH2 inhibitors treatment group are marked as * *p* < 0.05 and ** *p* < 0.01, respectively.

**Figure 5 biology-09-00093-f005:**
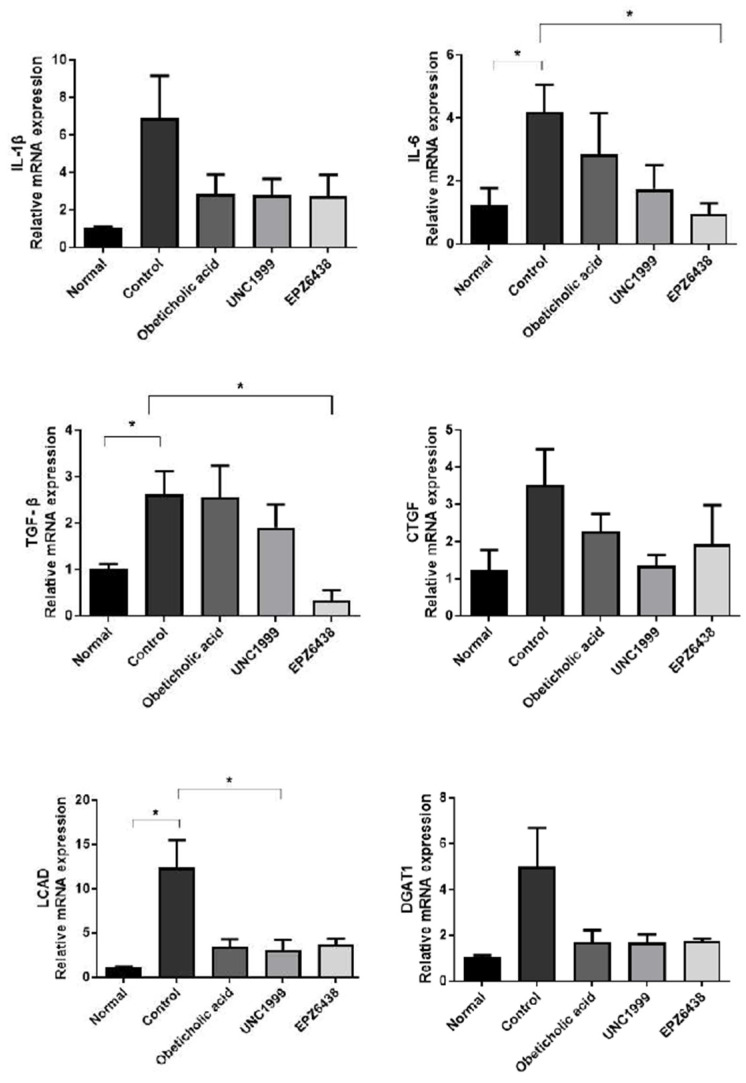
Treatment with EZH2 inhibitors inhibits liver inflammation and fibrosis-related genes in STAM NASH mice. Gene expression of inflammatory and fibrosis markers from Real-Time PCR. Expression of marker genes in healthy (Normal), STAM-Vehicle (Control), STAM-Obeticholic acid (Obeticholic acid), STAM-UNC1999 (UNC1999) and STAM-EZH2 (EPZ6438) groups in the NASH mice. The final concentration of Obeticholic acid, UNC1999 and EPZ6438 is 10 mg/kg. Statistically significant differences versus the STAM-Vehicle group and EZH2 inhibitors treatment group are marked as * *p* < 0.05, respectively.

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
