# Peer review of "The Role of the Histone Methyltransferase EZH2 in Liver Inflammation and Fibrosis in STAM NASH Mice"

_biology, 2020, doi:10.3390/biology9050093_

Round 1

Reviewer 1 Report

In this resubmitted manuscript, the authors have expanded on some of their initial observations to try to prove that EZH2 inhibitors may be useful in the STAM NASH model. However, the manuscript is still missing key controls and experiments and some of the data presented do not support the authors’ conclusions.

Major comment:

Given that EZH2 inhibition has no effect on glucose or Alt levels, and only UNC1999 while the more well characterized tazemetostat has no effect on TNF-alpha, I think that the authors’ claim that EZH2 inhibition is protective against inflammation is not supported.

Comments:

  1. The authors must expand upon the rationale as to why the treatment was started at 6 weeks rather than at 4 weeks to correspond with the start of the high-fat diet
  2. Figure 1: The authors do not describe how many mice they used in each of the five groups and for the H&E and other stains. Also, there should be some quantification in addition to the representative images to support their claims.
  3. For figure 1: the H&E and Oil red stains should be more descriptive to orient the reader as to what is really being measured.
  4. Line 182: “Although the reduction was not statistically significant, the level of ALT and glucose in STAM mice treated with EZH2 inhibitors was similar to those in STAM mice controls” this is a misleading statement. The data shows that there is no effect on glucose or ALT levels using EZH2 inhibitors.
  5. Figure 3: Based on the data shown in this figure (no change in 3A using tazemetostat, and no changes in 3B using either UNC or EPZ compounds), the authors cannot claim that EZH2 inhibition has a protective effect on liver inflammation. The data presented can only be interpreted as having no effect.
  6. Figure 3A: how does obeticholic acid treatment affect TNF-alpha? The positive control should be depicted in this experiment
  7. How does EZH2 inhibition change insulin, cholesterol and triglyceride levels? If EZH2 inhibition does indeed play a protective role in NASH mice, having this information is critical.
  8. Figure 4: given that UNC1999 and Tazemetostat are catalytic inhibitors of EZH2, I don’t understand why EZH2 mRNA is decreased upon treatment with these inhibitors. The authors must either present an explanation with references.
  9. Figure 4 is also missing the obeticholic acid control. Also, it is a little weird that only one direct EZH2 gene is measured using RT-PCR.
  10. A previous paper published in 2013 (Vella et al, 2013, IJMS) found that depletion of EZH2 exacerbates inflammation in NAFLD. While I recognize that the mouse models are different, the authors must address this in the discussion in more detail.

Author Response

1. The authors must expand upon the rationale as to why the treatment was started at 6 weeks rather than at 4 weeks to correspond with the start of the high-fat diet

In 2019 paper of STAM mice (Exp Anim 2019, 68, 417-428), histological analysis evaluated that STAM mice had liver steatosis, lobular inflammation, hepatocyte ballooning at 6 weeks and had fibrosis at 10 weeks with the start of the high-fat diet at 4 weeks. We can choose 6 weeks of mice as a starting point of NASH on NASH treatment concept.

2. Figure 1: The authors do not describe how many mice they used in each of the five groups and for the H&E and other stains. Also, there should be some quantification in addition to the representative images to support their claims.

We added detail information of many mice of the five groups for treatment of NASH and fixed Figure 1 legend for representative images of H&E and Oil red stains.

3. For figure 1: the H&E and Oil red stains should be more descriptive to orient the reader as to what is really being measured.

Based on reviewer’s comments, we re-wrote manuscripts in Results and figure legend section.

4. Line 182: “Although the reduction was not statistically significant, the level of ALT and glucose in STAM mice treated with EZH2 inhibitors was similar to those in STAM mice controls” this is a misleading statement. The data shows that there is no effect on glucose or ALT levels using EZH2 inhibitors.

Based on reviewer’s comments, we modified wrong statements (no effect on glucose or ALT levels after treatment with EZH2 inhibitors) and re-wrote manuscripts in Results section.

5. Figure 3: Based on the data shown in this figure (no change in 3A using tazemetostat, and no changes in 3B using either UNC or EPZ compounds), the authors cannot claim that EZH2 inhibition has a protective effect on liver inflammation. The data presented can only be interpreted as having no effect.

Based on reviewer’s comments, we modified wrong statements (EZH2 inhibition has a protective effect on liver inflammation) and re-wrote manuscripts in Results section.

6. Figure 3A: how does obeticholic acid treatment affect TNF-alpha? The positive control should be depicted in this experiment

As you know, we can show the results of ALT and glucose through serum test. However, we could not obtain the results of TNF of full experiment for five groups because of lack of serum samples.

7. How does EZH2 inhibition change insulin, cholesterol and triglyceride levels? If EZH2 inhibition does indeed play a protective role in NASH mice, having this information is critical.

Based on reviewer’s comments, we had data of triglyceride levels test in serum. However, there was no change the level of triglyceride between STAM mice treated with EZH2 inhibitors and STAM mice controls.

8. Figure 4: given that UNC1999 and Tazemetostat are catalytic inhibitors of EZH2, I don’t understand why EZH2 mRNA is decreased upon treatment with these inhibitors. The authors must either present an explanation with references.

Based on reviewer’s comments, we found the references to explain situation of decreasing EZH2 mRNA with these inhibitors. In lung fibroblast LL29 and chondrosarcoma cells, treatment of 3-Deazaneplanosin A (DZNep), EZH2 inhibitor reduce EZH2 protein expression. We think that EZH2 inhibitor regulates expression through other drug signaling pathway.

9. Figure 4 is also missing the obeticholic acid control. Also, it is a little weird that only one direct EZH2 gene is measured using RT-PCR.

We aimed to determine whether the EZH2 inhibition with inhibitors on NASH mice may be directly regulated by EZH2 target genes changes in mouse tissues. To explore this hypothesis, we tested only small molecule EZH2 inhibitors, UNC1999 and EPZ6438 except obeticholic acid. We think that treatment of obeticholic acid could not explain the direct regulation of EZH2 signaling pathway.

10. A previous paper published in 2013 (Vella et al, 2013, IJMS) found that depletion of EZH2 exacerbates inflammation in NAFLD. While I recognize that the mouse models are different, the authors must address this in the discussion in more detail.

A previous paper published in 2013 (Vella et al, 2013, IJMS) show that EZH2 expression decreased in high-fat/high-fructose diet (HFa/HFr-D) in vivo NAFLD model and palmitic/oleic acid (PA/OA) in vitro lipid accumulation HepG2 model. Treatment of DZNep as EZH2 inhibitor induced lipid accumulation in PA/OA induce in vitro HepG2 model. However, we suggested that opposite effect of EZH2 expression and EZH2 inhibitor might cause NASH model difference between HFa/HFr-D induced NAFLD and streptozotocin (STZ)/high-fat induced NASH model.

Reviewer 2 Report

The authors have now added more data to support their claims. 

However, a few minor corrections are still required:

1) Fig 1A is currently not cited in the text.

2) Fig 5: The color code/filling of the bars needs to be consistent across the different genes (for example "Normal" is sometimes white, sometimes filled).

3) Figs 4 and 5: For the statistical significances, the authors need to show with lines on the figures which conditions have been compared. It is unclear which comparisons have been performed (EZH2 inhibitor vs "Normal" or vs "Control" is not giving the same information).

Author Response

1. Fig 1A is currently not cited in the text.

Based on reviewer’s comments, we re-wrote manuscripts in Results and figure legend section.

2. Fig 5: The color code/filling of the bars needs to be consistent across the different genes (for example "Normal" is sometimes white, sometimes filled).

Based on reviewer’s comments, we modified it and added the Figure 5.

3. Figs 4 and 5: For the statistical significances, the authors need to show with lines on the figures which conditions have been compared. It is unclear which comparisons have been performed (EZH2 inhibitor vs "Normal" or vs "Control" is not giving the same information).

Based on reviewer’s comments, we rearranged the data including statistical significances and added the Figure 4, 5.

Reviewer 3 Report

The revised manuscript is definitely an improvement over the first submission, however the following points need to be addressed: 

I am confused regarding the rearrangement of the data in Figure 3B (originally Figure 2B). Based on the original submission all of the ALT values for the control and EZH2  inhibitors had U/L values < 100, however in the reformatted Figure 3B these U/L values (except normal) are now reported >> 100 U/L.  Please clear up this inconsistency.   The p-values are not marked on these graphs as mentioned in the figure legend.  Please mark the p-values on the graphs.

On line 64 please put (STZ) after streptozotocin as this acronym was used in Figure 1A.  

Figure 5. The labels for 3 of the graphs on the x-axis have been moved on-top of the graphs. Please get these labels all back below the x-axis. For these 3 mis-formatted graphs, no p-values have been marked. Please add the appropriate p values. In the legend of Figure 5 **p<0.01 was listed, however none of the graphs in Figure 5 have been marked with this p value. Please correct this inconsistency.

Lines 246-247. The sentence “We suggest that NASH model difference may be change the expression and mechanism of EZH2.” is poorly worded.  Please correct the grammar of this sentence and expand on what type of EZH2 expression and mechanism is being referred to. 

Author Response

1. I am confused regarding the rearrangement of the data in Figure 3B (originally Figure 2B). Based on the original submission all of the ALT values for the control and EZH2 inhibitors had U/L values < 100, however in the reformatted Figure 3B these U/L values (except normal) are now reported >> 100 U/L.  Please clear up this inconsistency. The p-values are not marked on these graphs as mentioned in the figure legend. Please mark the p-values on the graphs.

Based on reviewer’s comments, we performed ALT values test with normal mice. In new data, we could obtain the results of ALT values of full experiment for five groups. The ALT values for the control and EZH2 inhibitors had U/L values < 100. We think that test condition (test kit, serum condition) would be little changed. In 1st, 2st data, there was no change the ALT values between STAM mice treated with EZH2 inhibitors and STAM mice controls. So, we can’t put the p-values on these graphs.

2. On line 64 please put (STZ) after streptozotocin as this acronym was used in Figure 1A. 

We modified it and re-wrote manuscripts in Introduction section

3. Figure 5. The labels for 3 of the graphs on the x-axis have been moved on-top of the graphs. Please get these labels all back below the x-axis. For these 3 mis-formatted graphs, no p-values have been marked. Please add the appropriate p values. In the legend of Figure 5 **p<0.01 was listed, however none of the graphs in Figure 5 have been marked with this p value. Please correct this inconsistency.

Based on reviewer’s comments, we modified it and added the Figure 5.

4. Lines 246-247. The sentence “We suggest that NASH model difference may be change the expression and mechanism of EZH2.” is poorly worded.  Please correct the grammar of this sentence and expand on what type of EZH2 expression and mechanism is being referred to.

We modified it and re-wrote manuscripts in result section

Round 2

Reviewer 1 Report

The authors have clarified the statements made about the effect of EZH2 inhibition on liver inflammation.

This manuscript is a resubmission of an earlier submission. The following is a list of the peer review reports and author responses from that submission.

Round 1

Reviewer 1 Report

In this manuscript, the authors address the potential use of EZH2 inhibitors in non-alcoholic steatohepatitis. Given that EZH2 inhibitors have recently been approved by the FDA and thee lack of therapeutic options for NASH, this topic will be of potential interest to a broad audience. However, there are several issues which must be addressed before the manuscript can be considered for publication- in particular, the claims made by the authors require more data than is presented in this manuscript in order to be supported.

Comments:

The introduction does not give sufficient background. For example, what is the STAM NASH mouse model? How is it different compared to other mouse models for NASH? What are the current options for treatment of NASH? There is not sufficient rationale presented to warrant this particular study and how it would add to the field. lines 59-61: what do the authors mean by this statement? “epigenetic regulation mechanisms involved in the progression of NASH are not the main factors affecting EZH2 and NASH. Therefore, we investigated the role of EZH2 in NASH”? This makes it seem like there was no rationale to testing EZH2 inhibitors in this context The methods are not detailed: when were the treatments administered? (the authors address this in section 3.2 but it should be specified earlier). Also, what is the rationale to why was the treatment was started at 6 weeks? Statistical analysis: the p values are not indicated on any of the graphs. Line 128: the authors state “we aimed to determine whether the protective effect of EZH2 inhibitors on liver inflammation may be explained…”. This statement makes it seem like there is data that already supports a protective role- however there are no references, and there is no data presented before this point that support this statement, Line 137-138: “therefore the resultant effect was reduced liver inflammation”. There is not enough data to conclude this, and the data the authors refer to (fig 2A) does not support this statement. In fact, the authors see no effect on TNF-alpha after treatment with EPZ6438- this data is opposite to what the authors claim. Fig 2A: how does obeticholic acid treatment affect TNF-alpha? The positive control should be depicted in this experiment Fig 2A: statistics are missing. Fig 2B: The Y-axis needs to be labelled. Section 3.2: The authors claim is that EZH2 inhibition protects against liver inflammation. However, the presented data is neither sufficient nor supportive of this claim. The authors need to test other markers of inflammation such as IFN-gamma and TGF-beta in addition to H&E data for testing steatosis/lobular inflammation (such as those depicted in Middleton et al, 2018, Sci. Reports). How does EZH2 inhibition change glucose, insulin, cholesterol and triglyceride levels? If EZH2 inhibition does indeed play a protective role in NASH mice, having this information is critical. Figure 3: statistics are missing General comment: A previous paper published in 2013 (Vella et al, 2013, IJMS) found that depletion of EZH2 exacerbates inflammation in NAFLD. While I recognize that the mouse models are different, the authors must address this in the discussion.

Author Response

In this manuscript, the authors address the potential use of EZH2 inhibitors in non-alcoholic steatohepatitis. Given that EZH2 inhibitors have recently been approved by the FDA and thee lack of therapeutic options for NASH, this topic will be of potential interest to a broad audience. However, there are several issues which must be addressed before the manuscript can be considered for publication- in particular, the claims made by the authors require more data than is presented in this manuscript in order to be supported.

Based on reviewer’s major comments for requiring more data to prove the claims, we performed the H&E staining and oil red o staining for histology of liver of EZH2 inhibitor treatment and EZH2 direct regulation in mechanism data. We added the data in figure 1, figure 4 and re-wrote manuscripts.

Comments:

The introduction does not give sufficient background. For example, what is the STAM NASH mouse model?

We added detail information of STAM NASH mouse and re-wrote introduction section in manuscripts.

How is it different compared to other mouse models for NASH?

We added information of comparing to other mouse models and re-wrote introduction section in manuscripts.

What are the current options for treatment of NASH? There is not sufficient rationale presented to warrant this particular study and how it would add to the field.

We added detail information of current options for treatment of NASH and re-wrote introduction section in manuscripts.

lines 59-61: what do the authors mean by this statement? “epigenetic regulation mechanisms involved in the progression of NASH are not the main factors affecting EZH2 and NASH. Therefore, we investigated the role of EZH2 in NASH”? This makes it seem like there was no rationale to testing EZH2 inhibitors in this context

Based on reviewer’s comments, we modified wrong statements and re-wrote manuscripts in Introduction section

The methods are not detailed: when were the treatments administered? (the authors address this in section 3.2 but it should be specified earlier). Also, what is the rationale to why was the treatment was started at 6 weeks?

Based on reviewer’s comments, we made and added the NASH STAM mice experimental design information in Figure 1A and re-wrote manuscripts in methods section

The STATM mouse model progressed non-alcoholic steatohepatitis (NASH) from 6 weeks of age to 9 weeks of age. So, we decide to start at 6 weeks.

Statistical analysis: the p values are not indicated on any of the graphs.

Based on reviewer’s comments, we added modified the p values of graphs and re-wrote manuscripts in Legend section for figure.

Line 128: the authors state “we aimed to determine whether the protective effect of EZH2 inhibitors on liver inflammation may be explained…”. This statement makes it seem like there is data that already supports a protective role- however there are no references, and there is no data presented before this point that support this statement.

Based on reviewer’s comments, we modified wrong statements and re-wrote manuscripts in Results section.

Line 137-138: “therefore the resultant effect was reduced liver inflammation”. There is not enough data to conclude this, and the data the authors refer to (fig 2A) does not support this statement. In fact, the authors see no effect on TNF-alpha after treatment with EPZ6438- this data is opposite to what the authors claim. Fig 2A: how does obeticholic acid treatment affect TNF-alpha? The positive control should be depicted in this experiment Fig 2A: statistics are missing.

Based on reviewer’s comments, we modified wrong statements (no effect on TNF-alpha after treatment with EPZ6438) and re-wrote manuscripts in Results section.

Figure 2A added obeticholic acid treatment group and re-wrote manuscripts in Results section.

Fig 2B: The Y-axis needs to be labelled. Section 3.2: The authors claim is that EZH2 inhibition protects against liver inflammation. However, the presented data is neither sufficient nor supportive of this claim. The authors need to test other markers of inflammation such as IFN-gamma and TGF-beta in addition to H&E data for testing steatosis/lobular inflammation (such as those depicted in Middleton et al, 2018, Sci. Reports). How does EZH2 inhibition change glucose, insulin, cholesterol and triglyceride levels? If EZH2 inhibition does indeed play a protective role in NASH mice, having this information is critical. Figure 3: statistics are missing General comment: A previous paper published in 2013 (Vella et al, 2013, IJMS) found that depletion of EZH2 exacerbates inflammation in NAFLD. While I recognize that the mouse models are different, the authors must address this in the discussion.

Figure 2B added Y-axis label and re-wrote manuscripts in Results section.

Based on reviewer’s comments, we performed expression of IFN-gamma and TGF-beta as markers of inflammation using real-time PCR and H&E data and oil red o staining for testing steatosis/lobular inflammation and added Figure 1A, 1B and re-wrote manuscripts in Results section.

Also, we performed glucose level in serum and added Figure 3B and re-wrote manuscripts in Results section.

According to Vella et al, 2013, IJMS, depletion of EZH2 exacerbates inflammation in NAFLD. Based on reviewer’s opinion, we also found model difference. Vella et al, 2013, IJMS used High fat diet Rat NAFLD model and re-wrote manuscripts in Results section. We think that NASH model difference may be change the expression and mechanism of EZH2.

Reviewer 2 Report

Lee et al investigated whether the histone methyltransferase, EZH2, is involved in non-alcoholic steatohepatitis (NASH). The authors used two inhibitors of EZH2 and a mouse model of NASH to investigate the effects of inhibiting EZH2 on body and organ weights, TNFα and alanine aminotransferase production, and expression of some inflammatory and fibrosis related genes. 

The observations could be potentailly interesting but the data are too preliminary and key controls are missing from the manuscript. 

Comments:

1) The authors should test the expression level of EZH2 to see if its expression is affected in NASH vs non-NASH. This should indicate whether EZH2, compared to other histone methyltransferases, is a good target for potential therapeutic usage.

2) There is no data showing whether the mice treated with the inhibitors are better off than the mice without treatment, which is ultimately the important information. 

3) There is no control showing that EZH2 inhibiton is working.

4) It will be out of the scope of the manuscript but it is a bit unexpected that inhibiting a repressing factor (EZH2) leads to a decrease in expression of the genes tested. The effects observed by the authors is likely an indirect effect of EZH2 inhibition (global changes in histone methylation such as bivalent chromatin becoming H3K4me3 only, changes in gene expression...).

Figures 2 and 3: the stars for statistical significance are not shown.

Figure 2: Thre should be normally two controls, untreated NASH mice and them ice treated with Obeticholic acid (the positive control). It is unclear what Control in this figure refers to. The number of samples are also different (5 for the Control, 3 for UNC1999 and 4 for EPZ6438).

Lines 46-48: EZH2 is defined two times.

Lines 59-60: "Epigenetic regulation ... affecting EZH2 and NASH". This phrase does not make sense and need to be corrected (it is reading as NASH is not affecting NASH).

Lines 113-114 and 128-129: The phrases have to be rewritten. Currently, they read like affirmation and if it is the case, they require references. Nothing in the Introduction or the beginning of the Results links "EZH2 inhibition to liver fat deposition" (lines 113-114) or "a protective effect of EZH2 inhibitors on liver inflammation" (lines 128-129).

Lines 114-117: Repetition. The two phrases can be merged into one.

Lines 187-188: "Therapeutic interventions with ... of STAM mice". Overstatement. The authors only show that some genes involved in inflammation and fibrosis are less expressed upon EZH2 inhibition and some of them are still expressed at level higher than "Normal". To show this, the authors have to performed histology of the mice liver. 

Author Response

Review2

Non-alcoholic fatty liver disease (NAFLD) is a leading form of chronic liver disease with few biomarkers and treatment options currently available. Non-alcoholic steatohepatitis (NASH), a progressive disease of NAFLD may lead to fibrosis, cirrhosis, and hepatocellular carcinoma. Epigenetic modification can contribute to the progression of NAFLD causing non-alcoholic steatohepatitis (NASH), in which the exact role of epigenetics remains poorly understood. To identify potential therapeutics for NASH, we tested small-molecule inhibitors of the epigenetic target histone methyltransferase EZH2, Tazemetostat (EPZ-6438), and UNC1999 in the STAM NASH mice. The results demonstrate that treatment with EZH2 inhibitors decreased serum TNF-alpha in NASH. In this study, we investigated that inhibition of EZH2 reduced mRNA expression of inflammatory cytokines and fibrosis markers in NASH mice. In conclusion, these results suggest that EZH2 may present a promising therapeutic target in the treatment of NASH.

Lee et al investigated whether the histone methyltransferase, EZH2, is involved in non-alcoholic steatohepatitis (NASH). The authors used two inhibitors of EZH2 and a mouse model of NASH to investigate the effects of inhibiting EZH2 on body and organ weights, TNFα and alanine aminotransferase production, and expression of some inflammatory and fibrosis related genes. The observations could be potentially interesting but the data are too preliminary and key controls are missing from the manuscript.

Based on reviewer’s major comments for requiring more data to prove the claims, we performed the H&E staining and oil red o staining for histology of liver of EZH2 inhibitor treatment and EZH2 direct regulation in mechanism data. We added the data in figure 1, figure 4 and re-wrote manuscripts.

Comments:

1) The authors should test the expression level of EZH2 to see if its expression is affected in NASH vs non-NASH. This should indicate whether EZH2, compared to other histone methyltransferases, is a good target for potential therapeutic usage.

To check the expression of EZH2 between NASH and non-NASH, we performed real-time PCR analysis in figure 4 and re-wrote manuscripts. We found that expression of EZH2 increased in NASH group compared to non-NASH group.

2) There is no data showing whether the mice treated with the inhibitors are better off than the mice without treatment, which is ultimately the important information.

Based on reviewer’s major comments for requiring more data to prove the claims, we performed the H&E staining and oil red o staining for histology of liver of EZH2 inhibitor treatment and EZH2 direct regulation in mechanism data. We added the data in figure 1, figure 4 and re-wrote manuscripts.

3) There is no control showing that EZH2 inhibiton is working.

Based on reviewer’s major comments for requiring more data to prove the claims, we performed EZH2 direct regulation in mechanism data. We added the data in figure 4 and re-wrote manuscripts. We found that increasing EZH2 in NASH group reduced EZH2 expression and EZH2 target gene regulation in Figure 4.

4) It will be out of the scope of the manuscript but it is a bit unexpected that inhibiting a repressing factor (EZH2) leads to a decrease in expression of the genes tested. The effects observed by the authors is likely an indirect effect of EZH2 inhibition (global changes in histone methylation such as bivalent chromatin becoming H3K4me3 only, changes in gene expression...).

To check the direct effect of EZH2 inhibition, we performed by real-time PCR analysis for Runx3 target gene of EZH2 and added the data in Figure 4 and re-wrote the entire manuscripts.

Figures 2 and 3: the stars for statistical significance are not shown.

We added statistical significance in Figures and re-wrote manuscripts.

Figure 2: There should be normally two controls, untreated NASH mice and them ice treated with Obeticholic acid (the positive control). It is unclear what Control in this figure refers to. The number of samples are also different (5 for the Control, 3 for UNC1999 and 4 for EPZ6438).

In Figure 2B, we rearranged the data including untreated NASH mice (control) and NASH mice + obeticholic acid (positive control) as clinical trial III compound for NASH treatment groups. We added the Figure 2. We re-wrote manuscripts about control information.

Lines 46-48: EZH2 is defined two times.

We modified it and re-wrote manuscripts in Introduction section.

Lines 59-60: "Epigenetic regulation ... affecting EZH2 and NASH". This phrase does not make sense and need to be corrected (it is reading as NASH is not affecting NASH).

Based on reviewer’s comments, we modified wrong statements and re-wrote manuscripts in Introduction section.

Lines 113-114 and 128-129: The phrases have to be rewritten. Currently, they read like affirmation and if it is the case, they require references. Nothing in the Introduction or the beginning of the Results links "EZH2 inhibition to liver fat deposition" (lines 113-114) or "a protective effect of EZH2 inhibitors on liver inflammation" (lines 128-129).

Based on reviewer’s comments, we modified wrong statements (Lines 113-114 and 128-129) and re-wrote manuscripts in Introduction section.

Lines 114-117: Repetition. The two phrases can be merged into one.

Based on reviewer’s comments, we modified it and re-wrote manuscripts.

Lines 187-188: "Therapeutic interventions with ... of STAM mice". Overstatement. The authors only show that some genes involved in inflammation and fibrosis are less expressed upon EZH2 inhibition and some of them are still expressed at level higher than "Normal". To show this, the authors have to performed histology of the mice liver.

Based on reviewer’s comments, we modified Overstatement (Lines 187-188) and re-wrote manuscripts in Introduction section.

Based on reviewer’s comments, we performed the H&E stain for histology of mice liver. We added the data in figure 1 and re-wrote manuscripts.

Reviewer 3 Report

The Communication entitled “The role of the histone methyltransferase EZH2 in liver inflammation and fibrosis in STAM NASH mice” provided a novel correlation between the reduction of liver inflammation (and fibrosis) and the inhibition of EZH2 in NASH mice.  The manuscript was well presented.  

My main concern regarding this paper is the lack of clarity and information concerning the various controls used in the experiments reported herein. 

Line 75 mentions a positive control (obeticholic acid) that is described in the graphs shown in Figure 3.  What about a negative control?  Was another compound investigated as a negative control? 

Regarding Figure 3,  another “Control” was used in each of the 6 graphs.  The Figure 3 legend mentioned a STAM-Vehicle as the “Control”.  Please describe the STAM-vehicle (Control) in the methods section as it is not described elsewhere in this paper other than this figure legend.

Regarding Figure 1 (A and B). What control is being referenced? STAM-Vehicle, obeticholic acid, etc?   No precise information is provided in the results section or in the Figure 1 legend.

Lines 133-140 and Figure 2.  What control is referenced to in Figure 2? No information is provided in the Figure legend.  On line 133 obeticholic acid is mentioned as positive control. Is this positive control the same as the STAM mice (control) group mentioned on line 137 and is referred to in Figure 2 A and B?  Please clear up any confusion regarding the controls utilized in this manuscript. 

Author Response

Review3

Non-alcoholic fatty liver disease (NAFLD) is a leading form of chronic liver disease with few biomarkers and treatment options currently available. Non-alcoholic steatohepatitis (NASH), a progressive disease of NAFLD may lead to fibrosis, cirrhosis, and hepatocellular carcinoma. Epigenetic modification can contribute to the progression of NAFLD causing non-alcoholic steatohepatitis (NASH), in which the exact role of epigenetics remains poorly understood. To identify potential therapeutics for NASH, we tested small-molecule inhibitors of the epigenetic target histone methyltransferase EZH2, Tazemetostat (EPZ-6438), and UNC1999 in the STAM NASH mice. The results demonstrate that treatment with EZH2 inhibitors decreased serum TNF-alpha in NASH. In this study, we investigated that inhibition of EZH2 reduced mRNA expression of inflammatory cytokines and fibrosis markers in NASH mice. In conclusion, these results suggest that EZH2 may present a promising therapeutic target in the treatment of NASH.

The Communication entitled “The role of the histone methyltransferase EZH2 in liver inflammation and fibrosis in STAM NASH mice” provided a novel correlation between the reduction of liver inflammation (and fibrosis) and the inhibition of EZH2 in NASH mice.  The manuscript was well presented.

Based on reviewer’s major comments for requiring more data to prove the claims, we performed the H&E staining and oil red o staining for histology of liver of EZH2 inhibitor treatment and EZH2 direct regulation in mechanism data. We added the data in figure 1, figure 4 and re-wrote manuscripts.

My main concern regarding this paper is the lack of clarity and information concerning the various controls used in the experiments reported herein. Line 75 mentions a positive control (obeticholic acid) that is described in the graphs shown in Figure 3.  What about a negative control?  Was another compound investigated as a negative control?

In the NASH mouse model, we classified the 4 groups in in vivo study.

Normal groups are healthy mice. Control group are Nonalcoholic steatohepatitis (NASH) mice. To show the effect of inhibitor in Nonalcoholic steatohepatitis (NASH), we treated two EZH2 inhibitor and positive control (obeticholic acid) as clinical trial III inhibitor. Therefore, we would not consider negative control in this study.

Regarding Figure 3, another “Control” was used in each of the 6 graphs.  The Figure 3 legend mentioned a STAM-Vehicle as the “Control”.  Please describe the STAM-vehicle (Control) in the methods section as it is not described elsewhere in this paper other than this figure legend.

Based on reviewer’s comments, we described the STAM-vehicle (Control) in methods and results section.

Regarding Figure 1 (A and B). What control is being referenced? STAM-Vehicle, obeticholic acid, etc?   No precise information is provided in the results section or in the Figure 1 legend.

We rearranged the data including STAM-Vehicle and obeticholic acid groups and added the Figure 2.

Lines 133-140 and Figure 2.  What control is referenced to in Figure 2? No information is provided in the Figure legend.  On line 133 obeticholic acid is mentioned as positive control. Is this positive control the same as the STAM mice (control) group mentioned on line 137 and is referred to in Figure 2 A and B?  Please clear up any confusion regarding the controls utilized in this manuscript.

In Figure 2B, we rearranged the data and divided five groups including normal group (Healthy mice), control group (NASH STAM mice), obeticholic acid group (NASH STAM mice + obeticholic acid), UNC1999 group (NASH STAM mice + UNC1999 EZH2 inhibitor), EPZ6438 (NASH STAM mice + EPZ6438 EZH2 inhibitor). We re-wrote manuscripts in results section about control information.